# Multi-Frame Neural Scene Flow: Learning Bounds and Algorithms

## Abstract

Although Neural Scene Flow Prior (NSFP) and its variants have shown remarkable performance in large out-of-distribution autonomous driving, the underlying explanation for their generalization capabilities remains unclear. To this end, we analyze the generalization capabilities of NSFP via uniform stability and find that it exhibits a generalization bound, which is inversely proportional to the number of point clouds. These findings provide solid theoretical evidence to explain the effectiveness of NSFP in large-scale point cloud scene flow estimation tasks for the first time. To enhance practical scene understanding, we extend NSFP and propose a multi-frame neural scene flow (MNSF) scheme, which extracts temporal information across multiple frames. In this way, MNSF has better temporal consistency than NSFP. Moreover, we theoretically analyze its generalization abilities and demonstrate that it achieves a tight generalization bound with a convergence rate similar to NSFP. Extensive experimental results on large-scale autonomous driving Waymo Open and Argoverse datasets demonstrate that MNSF achieves state-of-the-art performance. *The code is attached to the submission.*

## 1 Introduction

Scene flow estimation stands out as a key endeavor for perception and understanding the 3D world in autonomous driving and robotics, aiming to determine motion fields within dynamic environments based on RGB images or point clouds (Teed & Deng, 2020; Liu et al., 2019b; Wang et al., 2022b). The existing point cloud scene flow consists of learning-based and optimization-based methods. Most learning-based methods (Liu et al., 2019b; Zhang et al., 2023; Peng et al., 2023) demonstrate superior performance on small-scale synthetic datasets (Menze & Geiger, 2015; Mayer et al., 2016), but struggle to generalize effectively to large-scale real-world scenarios (Chodosh et al., 2024; Li et al., 2023). In contrast, optimization-based methods (Li et al., 2021; 2023) show superior generalization performance in real-world autonomous driving scenarios, *e.g.,* Waymo and Argoverse (Sun et al., 2020; Chang et al., 2019).

As a classical optimization-based method, NSFP (Li et al., 2021) has demonstrated its strong capability to handle dense point clouds (about 150k+ points), showcasing remarkable generalization capabilities in open-world perception scenarios (Najibi et al., 2022; Chodosh et al., 2024). In addition, FNSF (Li et al., 2023) employs a distance transform strategy (Rosenfeld & Pfaltz, 1966; Breu et al., 1995) to significantly accelerate the optimization speed of NSFP without sacrificing its performance on out-of-distribution (OOD) autonomous driving scenes. Thus, NSFP and FNSF emerge as potentially powerful and dependable methods. However, the exceptional performance of NSFP and FNSF in processing large-scale point clouds needs theoretical analysis and still remains an intuition or empirical finding. The lack of a deeper understanding of NSFP hinders further progress in the field of neural scene flow estimation.

To address this issue, we conduct a theoretical investigation into the generalization error of NSFP through the framework of uniform stability (Bartlett & Mendelson, 2002; Bousquet & Elisseeff, 2002). Our findings reveal that the upper bound of NSFP's generalization error inversely correlates with the number of input point clouds. This analysis provides a foundational understanding of why NSFP excels in managing large-scale scene flow optimization tasks.

However, we can not further improve NSFP and FNSF by directly increasing the number of points, because the full point cloud in a frame (about 100k points) has already been used as the input. There-

fore, we seek to exploit the valuable information from previous frames from a temporal perspective, which improves the temporal consistency and overcomes the upper bound of point numbers in each frame. In this way, we aim to improve the scene flow estimation ($t \to t$+1) by using previous frames ($t$-1 $\to t$). Surprisingly, there appears to be a notable gap in research focused on utilizing such valuable temporal information for improving the two-frame point cloud scene flow estimations. Such a gap is particularly unexpected, because the extensive body of research in optical flow estimation (Wulff et al., 2017; Janai et al., 2018; Maurer & Bruhn, 2018; Liu et al., 2019a; Stone et al., 2021; Hur & Roth, 2021; Mehl et al., 2023) have shown the importance of temporal information from previous frames, even amidst rapid motion changes in optical flow.

In this paper, we propose a straightforward and efficient approach, namely MNSF, for estimating scene flow by using multiple frames. Specifically, we employ two FSNF models to calculate the forward ($t \to t$+1) and backward ($t \to t$-1) flows, respectively. These flows, naturally opposing in direction, are then reconciled through a motion model that inverts the backward flow. In this way, the inverted backward flow and forward flow are aligned in the same temporal direction. Furthermore, we introduce a temporal fusion module to encode these flows and predict the final flow. More crucially, we theoretically derive that the generalization error of MNSF is bounded, which guarantees the convergence of optimization. Experimental results on Waymo Open and Argoverse datasets show that MNSF outperforms FNSF by a large margin. We expect this study to provide analytical insights and encourage investigation into exploiting temporal information in scene flow estimation.

## 2 RELATED WORK

**Scene flow estimation.** Scene flow estimation from 2D images has been extensively explored in recent years (Teed & Deng, 2020; Menze & Geiger, 2015; Ma et al., 2019; Schuster et al., 2021; Maurer & Bruhn, 2018; Hur & Roth, 2021; Jiang et al., 2019). On the other hand, researchers estimate scene flow directly from 3D point clouds via full/self-supervised training schemes (Liu et al., 2019b;c; Gu et al., 2019; Wang et al., 2020; Puy et al., 2020; Kittenplon et al., 2021; Wang et al., 2021; Wu et al., 2020; Vedder et al., 2023; Wang et al., 2022b; Li et al., 2022; Zhang et al., 2023; Peng et al., 2023; Lang et al., 2023; Jiang et al., 2024). Specifically, these methods mainly extract point-based features and compute correspondences between two point clouds. Based on accurate correspondences, these methods achieve superior performance on synthetic KITTI Scene Flow (Menze & Geiger, 2015) and FlyingThings3D (Mayer et al., 2016) datasets. However, they fail to generalize to more realistic and larger autonomous driving scenarios (Pontes et al., 2020; Li et al., 2021; Najibi et al., 2022; Dong et al., 2022; Jin et al., 2022; Chodosh et al., 2023), *e.g.*, Waymo Open (Sun et al., 2020) and Argoverse (Chang et al., 2019) datasets. In comparison, NSFP (Li et al., 2021) uses a Multi-Layer Perception (MLP) to estimate the scene flow and demonstrates powerful generalization ability in large-scale autonomous driving scenarios. More recently, FNSF (Li et al., 2023) speeds up NSFP by using Distance Transform without sacrificing the performance.

**Multi-frame optical flow.** Extensive studies focus on using multi-frames to estimate optical flow (Golyanik et al., 2017; Maurer & Bruhn, 2018; Ren et al., 2019; Schuster et al., 2021; Hur & Roth, 2021; Mehl et al., 2023). Ren et al. (2019) discovers that performance improvements are relatively smaller when the frame number is more than three. In this way, these studies obtain more accurate results by considering three consecutive frames, which achieves a compromise between temporal information and efficiency (Wulff et al., 2017; Janai et al., 2018; Liu et al., 2019a; Stone et al., 2021). Specifically, these methods aim to learn a motion model across different frames, because optical flow fields are temporally smooth and distributed around a low-dimensional linear subspace (Irani, 1999; Janai et al., 2018). The motion model can exploit valuable information and predict the motion field of the current frame based on previous frames. Then, a fusion module combines the previous and current predictions to estimate an accurate result in the current frame.

## 3 APPROACH

**Preliminary: Two-frame point cloud scene flow optimization.** Let $\mathcal{S}_1$ and $\mathcal{S}_2$ denote the 3D point cloud sampled from a dynamic scene at time $t$-1 and $t$, respectively. Due to the movement and occlusion, the number of points in $\mathcal{S}_1$ and $\mathcal{S}_2$ are different and not in correspondence, *i.e.*, $|\mathcal{S}_1| \neq |\mathcal{S}_2|$. Let $\mathbf{f} \in \mathbb{R}^3$ denote a translational vector (flow vector) of a point $\mathbf{p} \in \mathcal{S}_1$ moving from time $t$-1 to

time $t$, *i.e.*, $\mathbf{p}' = \mathbf{p} + \mathbf{f}$. The scene flow $\mathcal{F}_1 = \{\mathbf{f}_i\}_{i=1}^{|\mathcal{S}_1|}$ is the set of translational vectors for all 3D points in $\mathcal{S}_1$. The optimal scene flow $\mathcal{F}^*$ obtains the minimal distance between the two point clouds $\mathcal{S}_1$ and $\mathcal{S}_2$. Due to the non-rigidity motion field of the dynamic scene, the optimization of the scene flow is inherently unconstrained. To this end, a regularization term C is usually used to constrain the motion field, *e.g.*, Laplacian regularizer (Pontes et al., 2020; Zeng et al., 2019). The optimization of the scene flow is defined by

$$\mathcal{F}^* = \arg\min_{\mathcal{F}_1} \sum_{\mathbf{p} \in \mathcal{S}_1} \mathrm{D}\left(\mathbf{p} + \mathbf{f}, \mathcal{S}_2\right) + \lambda \mathrm{C}, \tag{1}$$

where D is a point distance function, *e.g.*, Chamfer distance (Fan et al., 2017). $\lambda$ is a the coefficient for the regularization term C.

**Neural scene flow prior.** NSFP employs traditional runtime optimization to determine the optimal weights for the neural network without relying on prior knowledge or human annotations. NSFP utilizes the loss function $L$, treating the architecture of the neural network as an implicit form of regularization, as follows:

$$L\left(\mathbf{\Theta}, \mathbf{p}; \mathcal{S}_2\right) = \arg\min_{\mathbf{\Theta}} \sum_{\mathbf{p} \in \mathcal{S}_1} \mathrm{D}\left(\mathbf{p} + g\left(\mathbf{p}; \mathbf{\Theta}\right), \mathcal{S}_2\right), \tag{2}$$

where $\mathbf{\Theta}$ denotes the weights of the neural network $g$. $\mathbf{p}$ is the input point cloud sampled at time $t$-1, and the flow vector $\mathbf{f} = g\left(\mathbf{p}; \mathbf{\Theta}\right)$ represents the output of the neural network $g$. In this way, $\mathbf{f}^* = g\left(\mathbf{p}; \mathbf{\Theta}^*\right)$ denotes the optimal flow vector. NSFP implements the neural network $g$ as an MLP and uses Chamfer distance as the loss function to optimize the scene flow.

### 3.1 THEORETICAL EXPLAINING THE GENERALIZATION ABILITY OF NSFP

Despite NSFP demonstrating astonishing generalization ability, it lacks a foundational theoretical analysis for its underlying mechanism. This mystery impedes the reliability and development of the neural scene flow area. In this section, we explore NSFP in-depth and present a detailed theoretical analysis based on uniform stability, which is defined as follows.

**Definition 1.** *Given some algorithm $A$ and training data pairs $(x, y)$, its uniform stability $\beta$ exists with respect to (w.r.t.) its loss function $\ell$ and some domain $\mathcal{Z}$ if the flowing holds*

$$\forall S \in Z, \forall m \in \{1, \cdots, |S|\}, \forall z = (x, y), z_i^{'} \in \mathcal{Z}, |\ell\left(y, h_S\left(x\right)\right) - \ell\left(y, h_{S^m}\left(x\right)\right)| \leq \beta, \tag{3}$$

*where $h_S$ represents the hypothesis function output by a learning algorithm given the training sample $S$. Additionally, $S^m$ refers to a modified version of the training sample $S$, where the $i$-th example $z_i$ is substituted with an independent and identically distributed example $z_i^{'}$. We note here that $\ell\left(y, h_S\left(x\right)\right)$ and $\ell\left(y, h_{S^m}\left(x\right)\right)$ are related to the empirical and generalization errors of the algorithm $A$.*

We aim to determine bounds on the discrepancy between empirical and generalization errors for specific algorithms, *e.g.,* NSFP. To derive the theoretical results, we need some mild assumptions for the statistics of the point clouds and the related neural networks. The interested readers are referred to the works (Devroye & Wagner, 1979; Bousquet & Elisseeff, 2002; Zhang, 2002; Liu et al., 2016) for more applications of the related assumptions.

**Assumption 1.** *Finite point clouds and bounded neural network parameters: All considered point clouds, such as $\mathbf{P} \in \mathcal{S}_1$ and $\mathbf{Q} \in \mathcal{S}_2$, contain a finite number of points ($|\mathcal{S}_i|$), and the vector spaces of both the point clouds and the neural network parameters ($\mathbf{\Theta}$) are bounded:*

$$|\mathcal{S}_i|_{i=1,2} < \infty, \|\mathbf{P}\|_F \leq \sigma_P, \|\mathbf{Q}\|_F \leq \sigma_Q, \|\mathbf{\Theta}\|_F \leq \sigma_{\mathbf{\Theta}}. \tag{4}$$

In this assumption, we bound the norm of point clouds and related neural networks, which is reasonable and achievable in practice for point clouds without outliers (substantial value).

Next, to facilitate downstream theoretical analysis on the generalization bound of the NSFP, we can reformulate the loss function in Eq. (2) as

$$L\left(\mathbf{\Theta}, \mathbf{p}; \mathcal{S}_2\right) = L_p\left(\mathbf{\Theta}, \mathbf{p}; \hat{\mathbf{x}}_k\right) + L_q\left(\mathbf{\Theta}, \hat{\mathbf{p}}_l; \mathbf{q}_k\right), \tag{5}$$

where $L_p\left(\mathbf{\Theta}, \mathbf{p}; \hat{\mathbf{x}}_k\right) = \frac{1}{|\mathcal{S}_2|} \sum_{j=1}^{|\mathcal{S}_2|} \|\mathbf{\Theta}\mathbf{p}_j + \mathbf{p}_j - \hat{\mathbf{x}}_k\|_2^2$, and $L_q\left(\mathbf{\Theta}, \hat{\mathbf{p}}_l; \mathbf{q}_k\right) = \frac{1}{|\mathcal{S}_3|} \sum_{k=1}^{|\mathcal{S}_3|} \|(\mathbf{\Theta}\hat{\mathbf{p}}_l + \hat{\mathbf{p}}_l) - \mathbf{q}_k\|_2^2$

with the minimum of summation operators being defined by

$$\hat{\mathbf{x}}_k = \arg\min_{\mathbf{x} \in \mathcal{S}_3} \|\mathbf{p} - \mathbf{x}\|_2^2 \,; \hat{\mathbf{p}}_l = \arg\min_{\mathbf{y} \in \mathcal{S}_2} \|\mathbf{q} - \mathbf{y}\|_2^2 = \arg\min_{\mathbf{p} \in \mathcal{S}_2} \|\mathbf{q} - (\mathbf{\Theta}\mathbf{p} + \mathbf{p})\|_2^2. \tag{6}$$

We have the following mild assumptions for the loss functions $L_p$ and $L_q$:

**Assumption 2.** *Bounded Loss Functions: For some $\sigma_p$, for any $\Theta, \Theta_m \in \Theta$, the loss function $L_p$ is bounded by*

$$|L_p(\Theta, \mathbf{p}; \hat{\mathbf{x}}_k) - L_p(\Theta_m, \mathbf{p}; \hat{\mathbf{x}}_k)| \le \sigma_P \|(\Theta - \Theta_m)\mathbf{p}\|_2. \tag{7}$$

*For any network outputs $\Theta\hat{\mathbf{p}}_k + \hat{\mathbf{p}}_k$ and $\Theta\hat{\mathbf{p}}_l + \hat{\mathbf{p}}_l$, the loss $L_q$ is $\sigma_\Theta + 1$ admissible, such that*

$$|L_q(\Theta, \tilde{\mathbf{p}}_l; \mathbf{q}_k) - L_q(\Theta_m, \hat{\mathbf{p}}_l; \mathbf{q}_k)| \le (\sigma_\Theta + 1)\|\tilde{\mathbf{p}}_l - \hat{\mathbf{p}}_l\|_2 \tag{8}$$

*Besides, $L_q$ is $c$ -strongly convex:*

$$\langle \tilde{\mathbf{p}}_l - \hat{\mathbf{p}}_l, \nabla L_q(., \tilde{\mathbf{p}}_l) - \nabla L_q(., \hat{\mathbf{p}}_l)\rangle \ge c\|\tilde{\mathbf{p}}_l - \hat{\mathbf{p}}_l\|_2^2. \tag{9}$$

The above assumption has been made or adopted across various scientific fields (Zhang, 2002; Liu et al., 2016). In our case, it is employed to establish an upper bound on network loss functions. The bounded $L_p$, as described in Eq. (7), addresses scenarios where training remains stable and no outliers exist in either the network or the point cloud. The assumptions for $L_q$ ensure that $q$ is optimally selected based on an estimate $\hat{\mathbf{p}}$, which is reasonable, as once the forward flow is optimized, this selection becomes static, and identifying the best candidate $q$ in $\mathcal{S}_3$ is optimal.

**Assumption 3.** *Bounded reconstruction from point cloud subset: There exists a subset $\Omega = \{\mathbf{d}_1, \cdots, \mathbf{d}_{|\Omega|}\} \subset \{\mathbf{p}_1, \cdots, \mathbf{p}_{|\mathcal{S}_2|}\}$ such that for any point cloud $\mathbf{p}$ in considered tasks, $\mathbf{p}$ can be reconstructed with a small error ($\|\eta\| \le \varepsilon$): $\mathbf{p} = \sum_{j=1}^{|\Omega|} \alpha_j \mathbf{d}_j + \eta_j$, where $\alpha \in R$ and $\|\alpha\| \le r$.*

We note that Assumption 3 is quite mild. For instance, if the feature space exhibits low-rank characteristics, which is common in point clouds, or if the data lies on a manifold, satisfying this assumption can be relatively straightforward, even in the context of conventional Chamfer distance estimation. Likewise, if the feature vectors are randomized, the assumption holds as long as the point cloud size approaches the dimensionality of the feature vector.

**Theorem 1.** *(Proof is in Appendix A) With the above definitions and some assumptions, for some random sample in $\{\mathcal{S}_2, \mathcal{S}_3\}$, with high probability, we have,*

$$\beta_{\text{NSFP}} \le \frac{|\Omega|\sigma_p}{4}\left(rv + \sqrt{r^2v^2 + \frac{8v\sigma_\Theta\varepsilon}{|\Omega|}}\right) + \sigma_\Theta\sigma_p\varepsilon, \tag{10}$$

*where $v = \frac{\sigma_p}{|\mathcal{S}_2|} + \frac{\sigma_\Theta + 1}{|\mathcal{S}_3|}$ and all variables except $\mathcal{S}_2$ and $\mathcal{S}_3$ can be considered as constants[1].*

**Remark 1.** *In our investigation of large-scale point cloud data analysis utilizing NSFP families, we are intrigued by the question of how enlarging the sample size influences its learning performance. Theorem 1 indicates that NSFP has a generalization bound with a fast convergence rate of order ($\mathcal{O}\left(\frac{1}{\sqrt{|\mathcal{S}_2|}} + \frac{1}{\sqrt{|\mathcal{S}_3|}}\right)$) with respect to its sample size $|\mathcal{S}_2|$ and $|\mathcal{S}_3|$. This theoretical result provides strong support for the superior performance of NSFP in the large-scale scene flow estimation (please see Tables 1 and 2), where $|\mathcal{S}_2| \to \infty$ and $|\mathcal{S}_3| \to \infty$.*

### 3.2 Multi-Frame Scene Flow Optimization

In this section, we propose MNSF as a simple and effective strategy for multi-frame point cloud scene flow estimation. Following previous multi-frame optical flow estimation methods (Wulff et al., 2017; Janai et al., 2018; Liu et al., 2019a; Stone et al., 2021), we consider three consecutive frames ($t$-1, $t$, and $t$+1) and aim to estimate the scene flow from frame $t$ to frame $t$+1. Specifically, let $\mathcal{S}_1$, $\mathcal{S}_2$, and $\mathcal{S}_3$ be three 3D point clouds sampled from a dynamic scene at time $t$-1, $t$, and $t$+1. The number of points in each point cloud, $|\mathcal{S}_1|$, $|\mathcal{S}_2|$, and $|\mathcal{S}_3|$, are typically different and not in correspondence, *i.e.*, $|\mathcal{S}_1| \ne |\mathcal{S}_2| \ne |\mathcal{S}_3|$.

Motion fields across different frames are temporally smooth (Irani, 1999; Janai et al., 2018), we aim to use motion fields in previous frames to improve the estimation of the scene flow in the current frame. Specifically, Figure 1 shows that two FSNF models are used to calculate the forward and backward flows, respectively. Then, a temporal inversion and a fusion module predict the final flow.

To effectively exploit temporal information from previous frames, we propose to use two models $g_f(\mathbf{p}; \Theta_\mathbf{f})$ and $g_b(\mathbf{p}; \Theta_\mathbf{b})$ to predict the forward scene flow $\mathcal{F}_2 = \{\mathbf{f}_i\}_{i=1}^{|\mathcal{S}_2|}$ ($t \to t$+1) and the backward scene flow $\mathcal{B}_2 = \{\mathbf{b}_i\}_{i=1}^{|\mathcal{S}_2|}$ ($t \to t$-1), respectively. The optimization of these two models can be

---

[1] Detailed definitions of these variables and constants are provided in the Appendix A.

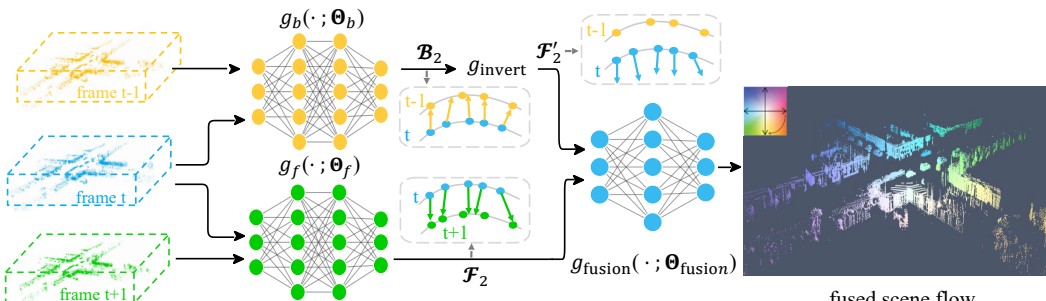

Figure 1: **Overview of the MNSF.** Given three consecutive frames ($t$-1, $t$, and $t$+1), we aim to estimate the scene flow from frame $t$ to frame $t$+1. Specifically, we use two models $g_f(\cdot;\mathbf{\Theta_f})$ and $g_b(\cdot;\mathbf{\Theta_b})$ to predict the forward scene flow $\mathcal{F}_2$ ($t\to t$+1) and the backward scene flow $\mathcal{B}_2$ ($t\to t$-1), respectively. Furthermore, a motion inverter $g_{\text{invert}}$ and a temporal fusion model $g_{\text{fusion}}(\cdot;\mathbf{\Theta}_{\text{fusion}})$ are used to estimate the fused scene flow. The upper left color wheel in the fused scene flow represents the flow magnitude and direction.

formulated as follows.

$$\mathbf{\Theta_f}^* = \arg\min_{\mathbf{\Theta_f}} \sum_{\mathbf{p}\in\mathcal{S}_2} \text{D}\left(\mathbf{p} + g_f\left(\mathbf{p};\mathbf{\Theta_f}\right),\mathcal{S}_3\right), \quad \mathbf{\Theta_b}^* = \arg\min_{\mathbf{\Theta_b}} \sum_{\mathbf{p}\in\mathcal{S}_2} \text{D}\left(\mathbf{p} + g_b\left(\mathbf{p};\mathbf{\Theta_b}\right),\mathcal{S}_1\right). \quad (11)$$

**Temporal scene flow inversion.** Given the forward and backward scene flow, we aim to further exploit useful temporal information from these flows. However, useful temporal information cannot be directly extracted, because the forward and the backward flow represent the opposite motion field, *i.e.*, $t\to t$+1 is opposite to $t\to t$-1. In this way, these flows conflict with each other. To this end, we introduce a motion model $g_{\text{invert}}(\mathbf{b};\mathbf{\Theta}_{\text{invert}})$ to invert the backward flow $\mathcal{B}_2 = \{\mathbf{b}_i\}_{i=1}^{|\mathcal{S}_2|}$ to the flow $\mathcal{F}_2' = \{\mathbf{f}_i'\}_{i=1}^{|\mathcal{S}_2|}$, which has the same direction of the forward flow. Therefore, we have $\mathbf{f}' = g_{\text{invert}}(\mathbf{b};\mathbf{\Theta}_{\text{invert}})$, where $\mathbf{b}\in\mathcal{B}_2$.

**Temporal fusion.** We can fuse the forward and the inverted backward scene flow and exploit useful temporal information. Specifically, we adopt an effective temporal fusion model $g_{\text{fusion}}\left(\mathbf{f},\mathbf{f}';\mathbf{\Theta}_{\text{fusion}}\right)$ to estimate the final scene flow, which is based on multi-frame point clouds. In this way, the fused flow can better overcome occlusions and out-of-view motion, because additional information on the occluded regions can be extracted from different frames/views (Maurer & Bruhn, 2018; Schuster et al., 2020; 2021).

$$\mathbf{\Theta}_{\text{invert}}^*, \mathbf{\Theta}_{\text{fusion}}^* = \arg\min_{\mathbf{\Theta}_{\text{invert}},\mathbf{\Theta}_{\text{fusion}}} \sum_{\mathbf{p}\in\mathcal{S}_2} \text{D}\left(\mathbf{p} + g_{\text{fusion}}\left(\mathbf{f},\mathbf{f}';\mathbf{\Theta}_{\text{fusion}}\right),\mathcal{S}_3\right), \quad (12)$$

where $\mathbf{f} = g_f(\mathbf{p};\mathbf{\Theta_f})$ and $\mathbf{f}' = g_{\text{invert}}(\mathbf{b};\mathbf{\Theta}_{\text{invert}})$.

Using a similar theoretical framework for the generalization analysis of the NSFP, we extend our analysis to the MNSF method in the subsequent sections.

**Theorem 2.** *(Proof is in Appendix A) Let* $\mathbf{\Theta}_{\text{fusion}} = \left[\mathbf{\Theta}_1^\top,\mathbf{\Theta}_2^\top\right]^\top$ *denote the parameters of the fusion model. For the proposed MNSF scheme, with high probability, its uniform stability ($\beta_{\text{MNSF}}$) is bounded by*

$$\beta_{\text{MNSF}} \leq \beta_{\text{NSFP}} + O\left(\frac{1}{|\mathcal{S}_2|}\right), \quad (13)$$

*where* $O\left(\frac{1}{|\mathcal{S}_2|}\right) = \frac{4\kappa^2\sigma_{\mathcal{S}_3}^2}{\lambda|\mathcal{S}_2|} + \left(\frac{8\kappa^2\sigma_{\mathcal{S}_3}^2}{\lambda} + 2\sigma_{\mathcal{S}_3}\right)\sqrt{\frac{\ln 1/\delta}{2|\mathcal{S}_2|}}$ *and* $\lambda = \frac{\|\mathbf{\Theta}_2\mathbf{\Theta}_b\|_2^2}{\|\mathbf{\Theta}_1\mathbf{\Theta}_f+\mathbf{I}\|_2^2}$. *Variables* $\kappa$, $\sigma_{\mathcal{S}_3}$, *and* $\delta$ *can be considered as constants.*

**Remark 2.** *Theorem 2 highlights two crucial properties of MNSF based on the loss function in Eq. (5): 1) The generalization bound of MNSF maintains a convergence rate comparable to that of NSFP, confirming that the incorporation of multiple frames in neural scene flow does not detract from convergence. 2) The upper bound of MNSF's generalization error aligns with that of NSFP as the size of $\mathcal{S}_2$ approaches infinity. This suggests that including the $t$-1 frame in the optimization preserves generalization. Further evidence supporting this claim can be found in the case study, and Tables 1 and 2.*

Table 1: **Evaluation on the Waymo Open Scene Flow dataset.** We follow previous studies (Li et al., 2021; 2023) to pre-process the Waymo Open dataset and generate 202 testing examples. Each point cloud contains 8k-144k points. The upper tabular between **blue bars** are evaluated with the full point cloud as the input, and the lower tabular between **orange bars** are evaluated with random samples 8,192 points as the input.

| Method | Supervision | Train set size | $\mathcal{E}(m)\downarrow$ | $Acc_5(\%)\uparrow$ | $Acc_{10}(\%)\uparrow$ | $Outliers(\%)\downarrow$ | $\theta_\epsilon(rad)\downarrow$ | $t\,(ms)\downarrow$ |
|---|---|---|---|---|---|---|---|---|
| NSFP (Li et al., 2021) | *Self* | 0 | 0.087 | 78.21 | 90.18 | 37.44 | 0.295 | 15310 |
| NSFP (linear) | *Self* | 0 | 0.153 | 60.28 | 75.89 | 53.19 | 0.353 | 7964 |
| FNSF | *Self* | 0 | 0.075 | 85.34 | 92.54 | 32.80 | 0.286 | 609 |
| FNSF (linear) | *Self* | 0 | 0.114 | 71.03 | 85.54 | 43.59 | 0.339 | **451** |
| FNSF (joint) | *Self* | 0 | 0.081 | 82.61 | 92.16 | 34.58 | 0.291 | 920 |
| FNSF (temporal encoding) | *Self* | 0 | 0.079 | 82.75 | 92.22 | 33.90 | 0.291 | 1011 |
| Ours (cycle consistency) | *Self* | 0 | 0.071 | 81.09 | 91.58 | 35.28 | 0.300 | 1831 |
| Ours | *Self* | 0 | **0.066** | **87.16** | **93.39** | **30.89** | **0.273** | 989 |
| FLOT (Puy et al., 2020) | *Full* | 18,000 | 0.694 | 2.62 | 11.89 | 94.74 | 0.792 | 133 |
| 3DFlow (Wang et al., 2022b) | *Full* | 18,000 | 2.088 | 1.60 | 4.92 | 98.94 | 1.845 | **80** |
| GMSF (Zhang et al., 2023) | *Full* | 18,000 | 8.058 | 0.00 | 0.01 | 99.96 | 1.341 | 245 |
| SCOOP (Lang et al., 2023) | *Self* | 1,800 | 0.313 | 41.86 | 65.02 | 64.71 | 0.474 | 558 |
| NSFP (Li et al., 2021) (8,192 pts) | *Self* | 0 | 0.109 | 64.63 | 81.82 | 45.60 | 0.338 | 4450 |
| FNSF (Li et al., 2023) (8,192 pts) | *Self* | 0 | 0.110 | 72.78 | 87.73 | 39.75 | 0.324 | 84 |
| Ours (8,192 pts) | *Self* | 0 | **0.102** | **79.42** | **90.87** | **36.51** | **0.321** | 160 |

Table 2: **Evaluation on the Argoverse Scene Flow dataset.** We pre-process the Argoverse dataset and generate 508 testing examples. Each point cloud contains 30k-70k points.

| Method | Supervision | Train set size | $\mathcal{E}(m)\downarrow$ | $Acc_5(\%)\uparrow$ | $Acc_{10}(\%)\uparrow$ | $Outliers(\%)\downarrow$ | $\theta_\epsilon(rad)\downarrow$ | $t\,(ms)\downarrow$ |
|---|---|---|---|---|---|---|---|---|
| NSFP (Li et al., 2021) | *Self* | 0 | 0.083 | 75.15 | 86.49 | 39.13 | 0.361 | 15214 |
| NSFP (linear) | *Self* | 0 | 0.107 | 58.39 | 76.39 | 55.21 | 0.337 | 2994 |
| FNSF | *Self* | 0 | 0.049 | 87.04 | 94.08 | 29.88 | 0.307 | 472 |
| FNSF (linear) | *Self* | 0 | 0.082 | 71.03 | 87.32 | 41.64 | 0.338 | **396** |
| FNSF (joint) | *Self* | 0 | 0.050 | 84.77 | 93.46 | 31.77 | 0.319 | 793 |
| FNSF (temporal encoding) | *Self* | 0 | 0.052 | 85.14 | 93.26 | 31.93 | 0.322 | 879 |
| Ours (cycle consistency) | *Self* | 0 | 0.054 | 83.26 | 92.36 | 32.81 | 0.325 | 1432 |
| Ours | *Self* | 0 | **0.044** | **88.75** | **94.83** | **28.86** | **0.299** | 851 |
| FLOT (Puy et al., 2020) | *Full* | 18,000 | 0.767 | 2.33 | 9.91 | 96.19 | 0.971 | 130 |
| 3DFlow (Wang et al., 2022b) | *Full* | 18,000 | 1.672 | 3.08 | 9.22 | 96.92 | 1.845 | **82** |
| GMSF (Zhang et al., 2023) | *Full* | 18,000 | 9.089 | 0.00 | 0.01 | 99.99 | 1.781 | 247 |
| SCOOP (Lang et al., 2023) | *Self* | 1,800 | 0.248 | 39.09 | 62.56 | 68.81 | 0.481 | 542 |
| NSFP (Li et al., 2021) (8,192 pts) | *Self* | 0 | 0.077 | 63.39 | 81.26 | 46.72 | 0.366 | 4390 |
| FNSF (Li et al., 2023) (8,192 pts) | *Self* | 0 | 0.081 | 75.87 | 87.85 | 39.10 | 0.372 | 83 |
| Ours (8,192 pts) | *Self* | 0 | **0.069** | **82.10** | **92.93** | **32.86** | **0.344** | 157 |

## 4 EXPERIMENTS

In this section, we evaluate MNSF on large-scale and realistic autonomous driving scenes. Specifically, we first introduce datasets and evaluation metrics. Then, we compare the proposed method with NSFP, FNSF, and different learning-based methods. Finally, we verify the effectiveness of each component in the proposed method with an ablation study.

**Datasets.** In this study, we focus on large-scale and lidar-based autonomous driving scenes. To this end, we conduct experiments on the Waymo Open (Sun et al., 2020) and the Argoverse (Chang et al., 2019) datasets. Specifically, we follow previous studies (Li et al., 2021; 2023) to pre-process these two open-world datasets and generate the pseudo ground truth scene flow. Please see more discussions in Appendix A.3.

**Metrics.** We evaluate the performance of the scene flow estimation based on widely used metrics from (Wu et al., 2020; Pontes et al., 2020; Li et al., 2021; 2023). (1) 3D end-point error $\mathcal{E}(m)$ measures the mean absolute distance between the estimated scene flow and the pseudo ground truth scene flow; (2) Strict accuracy $Acc_5(\%)$ represents the ratio of points that the absolute point error $\mathcal{E} < 0.05$m or the relative point error $\mathcal{E}' < 0.05$; (3) Relaxed accuracy $Acc_{10}(\%)$ represents the ratio of points that the absolute point error $\mathcal{E} < 0.1$m or the relative point error $\mathcal{E}' < 0.1$; (4) Outlier $Outliers(\%)$ represents the ratio of points that the absolute point error $\mathcal{E} > 0.3$m or the relative point error $\mathcal{E}' > 0.1$. In this way, $Inliers = 1 - Outliers$; (5) Angle error $\theta_\epsilon(rad)$ measures the mean angle error between the estimated scene flow and the pseudo ground truth scene flow; (6) Inference time $t(ms)$ measures the computation time for the scene flow estimation.

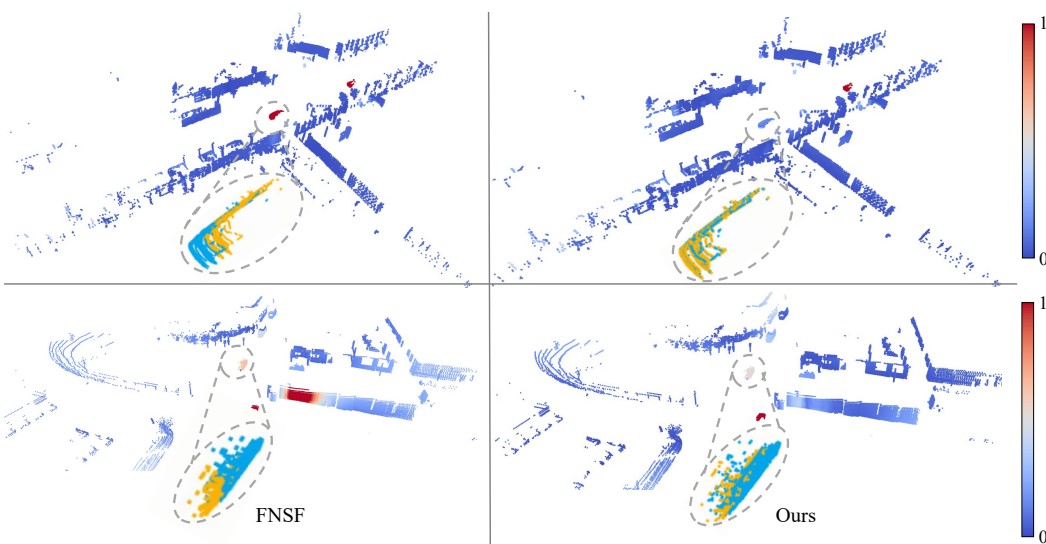

Figure 2: Visual comparison between FNSF and MNSF on the Argoverse dataset. For each point, color represents the normalized 3D end-point error $\mathcal{E}$. In this way, blue indicates the estimation of the flow is accurate. The detailed view demonstrates two point clouds aligned by the estimated flow.

**Implementation details.** We introduce details of implementation for each compared method. **(1) NSFP (Li et al., 2021).** We follow NSFP (Li et al., 2021) to use an 8-layer MLP, and the weights of the MLP are randomly initialized before optimizing each pair of point clouds; **(2) NSFP (linear).** Following (Li et al., 2023), we implement NSFP via 8 linear layers and compute the Kronecker product of the per-axis encoding; **(3) FNSF (Li et al., 2023).** For a fair comparison, we implement FNSF with an 8-layer MLP, and the grid cell size is 0.1 meters; **(4) FNSF (linear).** We also implement FNSF via a linear model with complex positional encodings. The settings of the linear model and positional encodings are the same as in **NSFP (linear)**; **(5) FNSF (joint).** To demonstrate the necessity of a dedicated strategy for utilizing temporal information, we use a single FNSF to jointly estimate the previous flow ($t$-1 $\rightarrow t$) and the current flow ($t \rightarrow t$+1); **(6) FNSF (temporal encoding).** Following (Zheng et al., 2023), we also use an FNSF to estimate the previous flow ($t$-1 $\rightarrow t$) and the current flow ($t \rightarrow t$+1) with temporal encoding. Please see more discussions in Appendix A.3; **(7) Ours.** We implement models $g_f$ and $g_b$ with 8-layer MLPs. These two models are independently trained. We simplify the model $g_{\text{invert}}$ as a constant model ($g_{\text{invert}}(\mathbf{b}) = -\mathbf{b}$) and adopt a 3-layer MLP as the fusion model $g_{\text{fusion}}$. The architecture of the fusion model is discussed in Section A.3. The grid cell size of FNSF is consistently set to 0.1 meters; **(8) Ours (cycle consistency).** We also implement the proposed method with a cycle consistency constraint in (Li et al., 2021), which aims to improve the smoothness of the scene flow estimation. Please see more discussions in Appendix A.3; **(9) FLOT (Puy et al., 2020), 3DFlow (Wang et al., 2022b), and GMSF (Zhang et al., 2023)** are supervised learning-based methods trained on the synthetic FlyingThings3D (Mayer et al., 2016) and the KITTI (Menze & Geiger, 2015) datasets. On the other hand, **SCOOP (Lang et al., 2023)** is a self-supervised method. These models are directly evaluated with pre-trained models and official codes released by the authors.

All experiments are conducted on a computer with a single NVIDIA RTX 3090Ti GPU and a Gen Intel (R) 24-Core (TM) i9-12900K CPU. We implement all compared models based on PyTorch.

## 4.1 COMPARISON OF PERFORMANCE

We evaluate and compare the proposed method with various state-of-the-art methods on the Waymo Open (Table 1) and the Argoverse (Table 2) datasets. For simplicity, we represent results on the Waymo Open (xx) and the Argoverse (yy) as xx/yy in the following paragraph. Figure 2 shows the visual comparison between FNSF and MNSF on the Argoverse dataset.

Table 3: **Performance of the proposed method with different components on the Waymo Open dataset.** All compared methods are evaluated with the full point cloud as the input.

| Model | Multi-frame | $g_{\text{invert}}$ | $g_{\text{fusion}}$ | $\mathcal{E} \downarrow$ | $Acc_5 \uparrow$ | $Acc_{10} \uparrow$ | $Outliers \downarrow$ | $\theta_\epsilon \downarrow$ | $t \downarrow$ |
|---|---|---|---|---|---|---|---|---|---|
| FNSF | | | | 0.075 | 85.34 | 92.54 | 32.80 | 0.286 | **609** |
| (a) | | | ✓ | 0.083 | 84.06 | 92.58 | 33.52 | 0.325 | 734 |
| (b) | ✓ | ✓ | | 0.070 | 82.94 | 92.64 | 32.89 | 0.284 | 613 |
| (c) | ✓ | | ✓ | 0.088 | 78.96 | 88.97 | 37.43 | 0.320 | 987 |
| (d) | ✓ | ✓ | ✓ | **0.066** | **87.16** | **93.39** | **30.89** | **0.273** | 989 |

**Dense scene flow estimation.** The ability to estimate dense scene flow is crucial, because each LiDAR scan often contains 100K - 1000K points in real-world autonomous driving scenarios (Jund et al., 2021). Therefore, we evaluate scene flow methods with the full point cloud as the input. NSFP achieves 78.21/75.15% strict accuracy, but the computation time costs 15310/15214 ms. To accelerate the optimization process, NSFP (linear) replaces the MLP with a linear model and positional encoding. In this way, NSFP (linear) speedups the optimization process almost two times and achieves worse performance compared to NSFP, *i.e.*, accuracy decreases by about 15%. FNSF achieves almost $30\times$ speedup and improves the strict accuracy to 85.34/87.04%. Meanwhile, FNSF (linear) slightly accelerates FNSF, suffering from a relatively large drop in performance.

All the above methods only use two frames ($t$ and $t$+1) and neglect to utilize previous frames. FNSF (joint) estimates the previous flow ($t$-1 $\rightarrow t$) and the current flow ($t \rightarrow t$+1) at the same time. However, such an intuitive scheme obtains worse strict accuracy (82.61/84.77%) than FNSF. The interpretation of this phenomenon is that a single MLP fails to encode different motion fields simultaneously, because points in the frame $t$-1 and the frame $t$ may have the same position $(x, y, z)$ with different motion fields. These inconsistent samples are difficult to be learned by DNNs (Liu et al., 2023). In contrast, the proposed method exploits valuable temporal information from previous frames and outperforms FNSF and FNSF (joint).

**OOD generalizability.** To conduct a fair comparison with learning-based methods (Puy et al., 2020; Wang et al., 2022b; Zhang et al., 2023; Lang et al., 2023), we further extend the proposed method to process a reduced number of points, *i.e.*, 8,192 points. Current learning-based methods could only process a fixed and small number of points due to their cumbersome networks (Peng et al., 2023; Zhang et al., 2023), *e.g.*, transformer-based architectures. To this end, these methods have to downsample or divide the entire lidar scan into smaller subsets/regions. Then, these learning-based methods can be iteratively used to predict the scene flow of each subset point cloud. In this way, such a compromising point cloud pre-process operation limits the generalization ability of learning-based methods on the large-scale OOD data and may lead to out-of-memory issues (Jund et al., 2021; Chodosh et al., 2023).

Table 1 and Table 2 show that supervised learning-based methods, including FLOT, 3DFlow, and GMSF, achieve limited performance on large-scale autonomous driving Waymo Open and Argoverse datasets. It is because of the huge domain shift between the training data and testing data (Pontes et al., 2020; Li et al., 2021; Najibi et al., 2022; Dong et al., 2022; Jin et al., 2022; Chodosh et al., 2023). In contrast, the self-supervised SCOOP outperforms its supervised counterparts and achieves 41.86/39.09% strict accuracy. However, the performance of SCOOP is still inferior to NSFP and FNSF. The proposed method outperforms FNSF by exploiting and utilizing temporal information from multi-frames. Although the computation cost of 3DFlow is the lowest among all compared methods, the proposed method achieves a balance between the performance and computational complexity. These experimental results and analysis indicate that the proposed method is robust for OOD data and is applicable to real-world autonomous driving scenarios.

**Discussions about learning-based methods.** Learning-based scene flow methods (Puy et al., 2020; Liu et al., 2019b;c; Wang et al., 2022b; Zhang et al., 2023; Peng et al., 2023) have exhibited remarkable speed and performance on small-scale synthetic datasets, *e.g.*, KITTI Scene Flow[2] (Menze & Geiger, 2015) and FlyingThings3D (Mayer et al., 2016) datasets. However, these methods heavily

---

[2]Point clouds in the KITTI dataset are limited to a specific range (35-meter within the scene center) with a small number of points (2048 or 8192 points).

Table 4: **Performance of fusion models with different depths on Waymo Open dataset.**

| Setting | $\mathcal{E}\downarrow$ | $Acc_5\uparrow$ | $Acc_{10}\uparrow$ | $\theta_\epsilon\downarrow$ |
|---|---|---|---|---|
| 2 layers | 0.069 | 86.84 | 93.07 | 0.286 |
| 3 layers | **0.066** | **87.16** | **93.39** | **0.273** |
| 5 layers | 0.068 | 86.33 | 93.16 | 0.281 |
| 7 layers | 0.107 | 83.50 | 92.30 | 0.303 |

Table 5: **Performance of different frame numbers on Waymo Open dataset.**

| Setting | $\mathcal{E}\downarrow$ | $Acc_5\uparrow$ | $Acc_{10}\uparrow$ | $\theta_\epsilon\downarrow$ |
|---|---|---|---|---|
| 2 frames | 0.083 | 84.46 | 92.58 | 0.313 |
| 3 frames | **0.066** | 87.16 | **93.39** | **0.273** |
| 4 frames | 0.070 | **87.64** | 93.38 | 0.279 |
| 5 frames | 0.085 | 87.48 | 93.31 | 0.291 |

rely on the high consistency between training scenarios and testing scenarios (Pontes et al., 2020; Li et al., 2021; Najibi et al., 2022; Dong et al., 2022; Jin et al., 2022; Chodosh et al., 2023), *e.g.*, viewpoints and coordinate systems. Thus, it is a challenge to use these learning-based methods in real-world applications, where training scenarios and testing scenarios are often inconsistent.

## 4.2 ABLATION STUDY AND CASE STUDY

In this section, we first conduct comprehensive experiments to verify the effectiveness of each component in the proposed method on the Waymo Open dataset. Specifically, given the forward and backward flows, the following four models are evaluated: (a) use the model $g_\text{fusion}$ to refine the forward flow; (b) use the model $g_\text{invert}$ to invert the backward flow, then directly compute the average of the inverted flow and the forward flow as the fused flow; (c) use the model $g_\text{fusion}$ to directly fuse the forward and backward flows; (d) equip all components, *i.e.*, MNSF.

Table 3 shows that each component is effective. Model (a) achieves comparable performance with FNSF without exploiting valuable information from previous frames. By coarsely using previous frames, model (b) slightly outperforms FNSF. Although model (c) uses information from previous frames, it performs worse than FNSF. This is because the forward and backward flows represent opposite directions and conflict with each other. Therefore, the direct fusion leads to performance degradation. Combining an inverter model $g_\text{invert}$ and a fusion model $g_\text{fusion}$ (*i.e.*, model (d)), achieves better performance than FNSF.

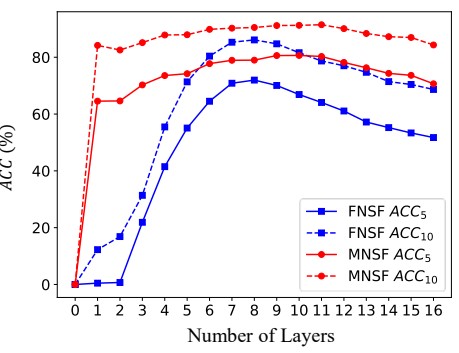

Figure 3: Performance of MNSF and FNSF with different number of layers on Waymo Open dataset.

**Scaling up model size.** To evaluate the performance of MNSF with increased model size, we demonstrate the scaling chart for MNSF and FNSF. Specifically, we increase the layer numbers of both MNSF and FNSF from 1 to 16 for a fair comparison. Figure 3 shows that MNSF achieves the best performance with a ten-layer MLP and FNSF with an eight-layer MLP. Increasing the layer number of FNSF does not further improve its performance when the layer number is larger than eight. Therefore, MNSF is more suitable to equip with deep MLPs and outperforms FNSF across different layer numbers.

**Depth of the temporal fusion model.** We illustrate the results of the proposed method with different depths of the temporal fusion model $g_\text{fusion}(\cdot\,\Theta_\text{fusion})$. Specifically, the temporal fusion model is set as 2-layer MLP, 3-layer MLP, 5-layer MLP, and 7-layer MLP. Table 4 shows that a 3-layer MLP temporal fusion model achieves the optimal performance. Therefore, a relatively small layer number of the temporal fusion model could better accomplish the fusion procedure.

**Number of frames.** We demonstrate the results of MNSF with different frame numbers. Table 5 shows that the multi-frame setting outperforms the 2-frame setting. It verifies that exploiting temporal information is useful for scene flow estimation. Table 5 indicates that the contribution of the temporal information is incremental, when the number of frames is larger than three. Such a finding is consistent with the previous work in the optical flow estimation (Ren et al., 2019) and object detection (Chen et al., 2022).

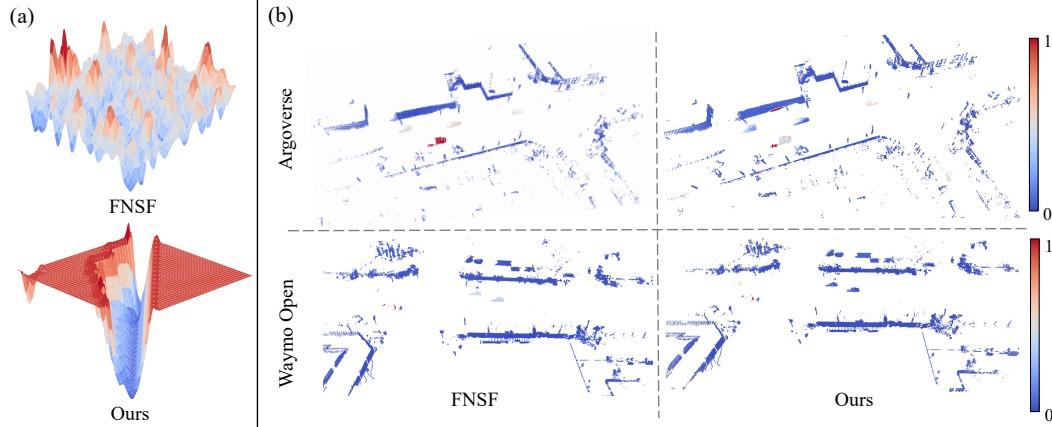

Figure 4: (a) The loss landscapes of FNSF and MNSF on the Argoverse dataset. Color represents the testing loss. MNSF eases the scene flow optimization process and has a more flat minimum. (b) Fast motion cases on the Argoverse and the Waymo Open datasets. Color represents the normalized 3D end-point error $\mathcal{E}$ for each point, and blue indicates the estimation of the flow is accurate.

**Loss landscape.** To further analyze the optimization difficulty of the neural scene flow estimation, we demonstrate the loss landscape of FNSF and MNSF in Figure 4(a). It is well known that the high flatness of the minima indicates good generalization ability (Li et al., 2018; Keskar et al., 2016; Ma et al., 2021; Chen et al., 2023). Figure 4(a) shows that the minima of MNSF are more flat than FNSF. Therefore, MNSF eases the scene flow optimization process and has better generalization ability, which also verifies the correctness of Theorem 2.

**Fast motion cases.** The ability to estimate dense scene flow of fast motion is important in real-world autonomous driving. Therefore, we demonstrate the error of the scene flow estimation in fast motion cases. Specifically, we select two fast motion cases from Argoverse and Waymo Open datasets based on the pseudo ground truth scene flow, respectively. Figure 4(b) shows that although the proposed method uses temporal information from previous frames, it can still accurately estimate the fast motion field. Such experimental results verify the robustness of MNSF in fast motion cases.

## 5 CONCLUSION

In this paper, we theoretically analyze NSFP's generalization ability and explain its effectiveness for large-scale point cloud scene flow estimation. Inspired by the theoretical findings, we propose an MNSF dedicated to large-scale point clouds. Furthermore, we conduct a theoretical analysis and demonstrate that MNSF's generalization error is bounded. Comprehensive case studies across five metrics confirm that MNSF significantly improves performance. Additionally, MNSF's robustness in processing fast motion cases and the high flatness of the minima in loss landscape underscore its effectiveness in large-scale OOD autonomous driving scenarios.

## 6 LIMITATION

MNSF needs to create a DT map using rasterization following (Li et al., 2023), which may bring discretization errors. In our case studies, we build a DT map with relatively fine-resolution grids, balancing the computation and the accuracy. In this way, the resolution of the DT map needs to be chosen for different scenarios. Moreover, from a theoretical standpoint, we present for the first time a generalization evaluation based on uniform stability for both NSFP and MNSF. We acknowledge that we do not verify the applicability of theoretical results for all methods, whether within or outside the NSFP families. We leave them in the future work.

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

# A  THEORETICAL ANALYSIS

Drawing on the theoretical frameworks proposed by (Devroye & Wagner, 1979; Bartlett & Mendelson, 2002; Bousquet & Elisseeff, 2002; Liu et al., 2016), we adopt uniform stability, as introduced by (Bartlett & Mendelson, 2002; Bousquet & Elisseeff, 2002), as a metric to evaluate the generalization performance of both NSFP and the method proposed in this study. We initiate by presenting the essential technical tools.

## A.1  NOTATIONS

Let $\mathcal{X} \in \mathbb{R}$ and $\mathcal{Y} \in \mathbb{R}$ be the input and output space, we consider the training dataset

$$\Phi = \left\{ z_1, \cdots, z_{|\Phi|} \right\}, \tag{1}$$

where we have $z_i = \{x_i, y_i\}|_{i=1,\cdots,|\Phi|}$ and $\mathcal{Z} = \mathcal{X} \times \mathcal{Y}$ drawn independent and identically distributed from some unknown distribution $\Xi$. The learning algorithm, denoted by $A$, is to learn some function from $\mathcal{Z}^{|\Phi|}$ into $\mathcal{F} \subset \mathcal{Y}^{\mathcal{X}}$, mapping the dataset $\Phi$ onto the function $A_\Phi$ from $\mathcal{X}$ to $\mathcal{Y}$. Since we are considering a neural network-based algorithm, $A$ here is related to the learnable neural network parameters. We use $\mathbb{E}_z$ to represent the expectation operator. Given a training dataset $\Phi$, we also consider a modified version by replacing the $i$-th element by a new sample $z_m'$, yielding

$$\Phi^m = \left\{ z_1, \cdots, z_{m-1}, z_m', , z_{m-1}, \cdots, z_{|\Phi|} \right\}. \tag{2}$$

We assume the replacement example $z_m'$ is drawn from $\Xi$ and is independent of $\Phi$. We use the $risk$ (also known as $generalization\ error$) to measure the performance of a learning algorithm (Bartlett & Mendelson, 2002; Bousquet & Elisseeff, 2002), which can be denoted by

$$R(A, \Phi) = \mathbb{E}_z \left[ \ell(A_\Phi, z) \right], \tag{3}$$

where $\ell$ represents the loss function of a learning algorithm. The classical estimator for the $risk$ of the dataset $\Phi^m$ is the $resubstitution\ estimate$ (also known as $empirical\ error$)(Bousquet & Elisseeff, 2002), defined as

$$R(A, \Phi^m) = \frac{1}{|\Phi|} \sum_{i=1}^{|\Phi|} \ell(A_{\Phi^m}, z_i). \tag{4}$$

## A.2  ASSUMPTIONS AND MAIN TOOLS

The objective of this study is to establish bounds on the disparity between empirical and generalization errors for particular algorithms, which can be defined in the following.

**Definition 1.** *Given some algorithm A, its uniform stability $\beta$ exists with respect to (w.r.t.) its loss function $\ell$ if the flowing holds*

$$\forall \Phi \in Z, \forall m \in \{1, \cdots, |\Phi|\},$$

$$\Delta R \stackrel{\Delta}{=} |R(A, \Phi) - R(A, \Phi^m)| \leq \beta. \tag{5}$$

To bound the uniform stability, we need some probability measure, such as the Bregman divergence (Mohri et al., 2018), which is defined by

**Definition 2. Bregman divergence:** *Let $L : \mathcal{H} \to \mathbb{R}$ be a strictly convex function that is continuously differentiable on int $\mathcal{H}$. For all distinct $g, h \in \mathcal{H}$, then the Bregman divergence is defined as*

$$B_L(g||h) = L(g) - L(h) - \langle g - h, \nabla L(h) \rangle \tag{6}$$

Some key properties of Bregman divergence (Mohri et al., 2018) are given in the following:

**Lemma 1.** *Bregman divergence is non-negative and additive. For example, give some convex functions $F_1$, $F_2$ and $F = F_1 + F_2$, for any $g, h \in \mathcal{H}$, we have*

$$B_F(g||h) = B_{F_1}(g||h) + B_{F_2}(g||h) \tag{7}$$

*and*

$$B_F(g||h) \geq 0. \tag{8}$$

To get the theoretical results, we need some mild assumptions for the statistics of the point clouds and the related neural networks. The interested readers are referred to the works (Devroye & Wagner, 1979; Bousquet & Elisseeff, 2002; Zhang, 2002; Liu et al., 2016) for more applications of the related assumptions.

**Assumption 1.** *Any point clouds ($\mathbf{P} \in S$) considered in the work contain a finite points and vector spaces of point clouds and neural network ($\mathbf{\Theta}$) are bounded,*

$$|\mathcal{S}_i|_{i=1,2,3} < \infty, \|\mathbf{P}\|_F \leq \sigma_P,$$

$$\|\mathbf{Q}\|_F \leq \sigma_Q, \|\mathbf{R}\|_F \leq \sigma_R, \|\mathbf{\Theta}\|_F \leq \sigma_{\mathbf{\Theta}}. \tag{9}$$

In this assumption, we bound the norm of point clouds and related neural networks (forward model), which is reasonable and achievable in practice for point clouds without outliers (substantial value).

To enable the downstream analysis without loss of generality, we assume the minimum of the summation operators are given by

$$\hat{\mathbf{x}}_k = \arg \min_{\mathbf{x} \in \mathcal{S}_3} \|\mathbf{p} - \mathbf{x}\|_2^2 \tag{10}$$

and

$$\hat{\mathbf{p}}_l = \arg \min_{\mathbf{y} \in \mathcal{S}_2} \|\mathbf{q} - \mathbf{y}\|_2^2 = \arg \min_{\mathbf{p} \in \mathcal{S}_2} \|\mathbf{q} - (\mathbf{\Theta}\mathbf{p} + \mathbf{p})\|_2^2. \tag{11}$$

Let $\mathbf{p}_i$ and $\mathbf{q}_j$ be the $i$-th and $j$-th point clouds in the $\mathcal{S}_2$ and $\mathcal{S}_3$, respectively. Then, for the NSFP problem, we can rewrite the corresponding loss function as

$$L(\mathbf{\Theta}, \mathbf{p}; \mathcal{S}_3) = L_p(\mathbf{\Theta}, \mathbf{p}; \hat{\mathbf{x}}_k) + L_q(\mathbf{\Theta}, \hat{\mathbf{p}}_l; \mathbf{q}_j), \tag{12}$$

where

$$L_p(\mathbf{\Theta}, \mathbf{p}; \hat{\mathbf{x}}_k) = \frac{1}{|\mathcal{S}_2|} \sum_{i=1}^{|\mathcal{S}_2|} \|\mathbf{\Theta}\mathbf{p}_i + \mathbf{p}_i - \hat{\mathbf{x}}_k\|_2^2,$$

and

$$L_q(\mathbf{\Theta}, \hat{\mathbf{p}}_l; \mathbf{q}) = \frac{1}{|\mathcal{S}_3|} \sum_{j=1}^{|\mathcal{S}_3|} \|(\mathbf{\Theta}\hat{\mathbf{p}}_l + \hat{\mathbf{p}}_l) - \mathbf{q}_j\|_2^2$$

We include the following mild assumptions for the loss functions $L_p$ and $L_q$:

**Assumption 2.** *For some $\sigma_p$, for any $\mathbf{\Theta}, \mathbf{\Theta}_m \in \Theta$, the loss function $L_p$ is bounded by*

$$|L_p(\mathbf{\Theta}, \mathbf{p}; \hat{\mathbf{x}}_k) - L_p(\mathbf{\Theta}_m, \mathbf{p}; \hat{\mathbf{x}}_k)| \leq \sigma_P \|(\mathbf{\Theta} - \mathbf{\Theta}_m)\mathbf{p}\|_2. \tag{13}$$

For any network outputs (estimates) $\mathbf{\Theta}\hat{\mathbf{p}}_k + \hat{\mathbf{p}}_k$ and $\mathbf{\Theta}\hat{\mathbf{p}}_l + \hat{\mathbf{p}}_l$, the loss $L_q$ is $\sigma_{\mathbf{\Theta}} + 1$ admissible, such that

$$|L_q(\mathbf{\Theta}, \tilde{\mathbf{p}}_l; \mathbf{q}_k) - L_q(\mathbf{\Theta}_m, \hat{\mathbf{p}}_l; \mathbf{q}_k)| \leq (\sigma_{\mathbf{\Theta}} + 1) \|\tilde{\mathbf{p}}_l - \hat{\mathbf{p}}_l\|_2 \tag{14}$$

Besides, $L_q$ is $c$-strongly convex:

$$\langle \tilde{\mathbf{p}}_l - \hat{\mathbf{p}}_l, \nabla L_q(., \tilde{\mathbf{p}}_l) - \nabla L_q(., \hat{\mathbf{p}}_l) \rangle \geq c \|\tilde{\mathbf{p}}_l - \hat{\mathbf{p}}_l\|_2^2. \tag{15}$$

**Assumption 3.** *There exists a subset $\Omega = \{\mathbf{d}_1, \cdots, \mathbf{d}_{|\Omega|}\} \subset \{\mathbf{p}_1, \cdots, \mathbf{p}_{|\mathcal{S}_2|}\}$ such that for any point cloud $\mathbf{p}$ in considered tasks, $\mathbf{p}$ can be reconstructed with a small reconstruction error ($\|\eta\| \leq \varepsilon$): $\mathbf{p} = \sum_{j=1}^{|\Omega|} \alpha_j \mathbf{d}_j + \eta_j$, where $\alpha \in R$ and $\|\alpha\| \leq r$.*

The above four assumptions were used to bound the network function, and similar assumptions have been used and demonstrated effective in theoretical works (Zhang, 2002; Liu et al., 2016). We begin our demonstration by presenting an outline of the proofs for our principal theories. We start by utilizing the statistical characteristics (specifically, Bregman convergence) of selected subset point clouds, constructing these subsets from the original point clouds. Subsequently, we delve into examining the upper bounds of these subset point clouds. The pivotal findings are then derived from this theoretical analysis and subsequent calculations.

### A.2.1 KEY THEOREMS

Our first goal here is to upper-bound the NSFP algorithm as defined in the following:

**Definition 3. Uniform Stability of NSFP:** *An algorithm is $\beta$ uniformly stable with respect to the loss function L if the following holds with high probability:*

$$\Delta R\left(L, \{\mathcal{S}2, \mathcal{S}_3\}\right) = |L_p\left(\Theta, \mathbf{p}; \hat{\mathbf{x}}_k\right) - L_p\left(\Theta_m, \mathbf{p}; \hat{\mathbf{x}}_k\right)| \leq \beta, \tag{16}$$

*where $\Theta_m$ is the optimal forward models of the loss function L over the datasets $S_2^m$ and $S_3^m$ in which we replace its m-th sample $(\mathbf{p}_m, \hat{\mathbf{p}}_l)$ by a random new point cloud $\left(\mathbf{p}'_m, \hat{\mathbf{p}}'_l\right)$.*

Based on the provided definitions, certain mild assumptions, and comprehensive derivations, we obtain the following theoretical theoretical results.

**Theorem 1.** *With the above definitions and some assumptions, for some random sample in $\{\mathcal{S}_2, \mathcal{S}_3\}$, with high probability, we have,*

$$\beta_{\mathrm{NSFP}} \leq \frac{|\Omega|\,\sigma_p}{4}\left(rv + \sqrt{r^2 v^2 + \frac{8v\sigma_\Theta \varepsilon}{|\Omega|}}\right) + \sigma_\Theta \sigma_p \varepsilon, \tag{17}$$

*where $v = \frac{\sigma_p}{|\mathcal{S}_2|} + \frac{\sigma_\Theta + 1}{|\mathcal{S}_3|}$ and all variables except $\mathcal{S}_2$ and $\mathcal{S}_3$ can be considered as constants.*

*Proof.* **Proof sketch:** To define limits on the differences between empirical errors and generalization errors for specific algorithms, we initially explore the statistical correlation between the subset and original point clouds. This exploration enables us to ascertain an upper limit for forward model errors. Subsequently, we focus on the Bregman divergence, utilizing it as a pivotal statistical metric, from which we deduce the crucial inequality. This process culminates in the formulation of a comprehensive proof of our theorems. It's important to mention that, although our analysis is based on a linear network model, empirical evidence from case studies has shown that it performs well in both linear and nonlinear network models.

**Statistical Relationship between the Subset and Original Point Clouds:** With Assumption 2 and Cauchy-Schwarz inequality, we have

$$\begin{aligned}
&|L_p\left(\Theta, \mathbf{p}; \hat{\mathbf{x}}_k\right) - L_p\left(\Theta_m, \mathbf{p}; \hat{\mathbf{x}}_k\right)| \\
&\leq \sigma_p \|(\Theta - \Theta_m)\,\mathbf{p}\|_2 \\
&\leq \sqrt{\sum_j \alpha_j^2} \sqrt{\sum_{j=1}^{|\Omega|} \|(\Theta - \Theta_m)\,\mathbf{d}_j\|_2^2} + \|(\Theta - \Theta_m)\|_2 \|\eta\|_2 \\
&\leq r\sqrt{\sum_{j=1}^{|\Omega|} \|(\Theta - \Theta_m)\,\mathbf{d}_j\|_2^2} + \frac{2\sigma_\Theta \varepsilon}{|\mathcal{S}_2|}
\end{aligned} \tag{18}$$

Then our goal is to bound the $\|(\Theta - \Theta_m)\,\mathbf{d}\|_2$, which is based on the Bregman divergence between the point clouds $\Phi$ and its subset $\Omega$.

With the definitions in Section A.1, we know that the loss function $L$ and $L_m$ are defined over the original dataset $\mathcal{S}_2$ and $\mathcal{S}_3$. For the same loss functions defined over the subset $\Omega$, we can denote them as $L^\Omega$ and $L_m^\Omega$ for notation compactness. Considering the non-negativity and additivity of the Bregman divergence (Lemma 1), we can have

$$B_{L_q}\left(\Theta_m \| \Theta\right) \leq B_L\left(\Theta_m \| \Theta\right), B_{L_q}\left(\Theta_m \| \Theta\right) \leq B_{L_m}\left(\Theta_m \| \Theta\right) \tag{19}$$

and

$$\begin{aligned}
&B_{L_q^\Omega}\left(\Theta_m \| \Theta\right) + B_{L_q^\Omega}\left(\Theta \| \Theta_m\right) \\
&\leq \kappa\left[B_{L_q}\left(\Theta_m \| \Theta\right) + B_{L_q}\left(\Theta \| \Theta_m\right)\right]
\end{aligned}, \tag{20}$$

for some $\kappa > 0$.

**Key Inequalities:** We concentrate on establishing the critical inequalities between the Bregman divergence of the initial point clouds and the divergence observed in their subsets. We start by

showing the key inequality of $B_{L_q^\Omega}\left(\boldsymbol{\Theta}_m||\boldsymbol{\Theta}\right) + B_{L_q^\Omega}\left(\boldsymbol{\Theta}||\boldsymbol{\Theta}_m\right)$:

$$
\begin{aligned}
& B_{L_q^\Omega}\left(\boldsymbol{\Theta}_m||\boldsymbol{\Theta}\right) + B_{L_q^\Omega}\left(\boldsymbol{\Theta}||\boldsymbol{\Theta}_m\right) \\
& = \frac{1}{|\Omega|}\sum_{i=1}^{|\Omega|}\left\langle \boldsymbol{\Theta} - \boldsymbol{\Theta}_m, \nabla L_q\left(\boldsymbol{\Theta}, \hat{\mathbf{p}}_l; \mathbf{q}_i\right)\mathbf{d}_i^T\right\rangle \\
& \quad - \frac{1}{|\Omega|}\sum_{i=1}^{|\Omega|}\left\langle \boldsymbol{\Theta} - \boldsymbol{\Theta}_m, \nabla L_q\left(\boldsymbol{\Theta}_m, \hat{\mathbf{p}}_l; \mathbf{q}_i\right)\mathbf{d}_i^T\right\rangle \\
& = \frac{1}{|\Omega|}\sum_{i=1}^{|\Omega|}\left\langle \left(\boldsymbol{\Theta} - \boldsymbol{\Theta}_m\right)\mathbf{d}_i, \nabla L_q\left(\boldsymbol{\Theta}, \hat{\mathbf{p}}_l; \mathbf{q}_i\right) - \nabla L_q\left(\boldsymbol{\Theta}_m, \hat{\mathbf{p}}_l; \mathbf{q}_i\right)\right\rangle \\
& \geq \frac{c}{|\Omega|}\sum_{i=1}^{|\Omega|}\left\|\left(\boldsymbol{\Theta} - \boldsymbol{\Theta}_m\right)\mathbf{d}_i\right\|_2^2
\end{aligned}
\tag{21}
$$

where the inequality holds from Assumptions 2 and results given in Eq. (19). Since the mean square error is considered, we have $c = 2$.

Since $\boldsymbol{\Theta}_m$ and $\boldsymbol{\Theta}$ are the optimal forward models of $L$ and $L_m$, we have $\nabla_L\left(\boldsymbol{\Theta}\right) = 0$ and $\nabla_{L_m}\left(\boldsymbol{\Theta}_m\right) = 0$. Then with the definition in Eq. (6), we obtain

$$
\begin{aligned}
& B_L\left(\boldsymbol{\Theta}_m||\boldsymbol{\Theta}\right) + B_{L_m}\left(\boldsymbol{\Theta}||\boldsymbol{\Theta}_m\right) \\
& = L\left(\boldsymbol{\Theta}_m\right) - L\left(\boldsymbol{\Theta}\right) + L_m\left(\boldsymbol{\Theta}\right) - L_m\left(\boldsymbol{\Theta}_m\right) \\
& = \left(L\left(\boldsymbol{\Theta}_m\right) - L_m\boldsymbol{\Theta}_m\right) + \left(L_m\left(\boldsymbol{\Theta}\right) - L\left(\boldsymbol{\Theta}\right)\right) \\
& = \frac{1}{|\mathcal{S}_2|}\left[L_p\left(\boldsymbol{\Theta}, \mathbf{p}_m; \hat{\mathbf{x}}_k\right) - L_p\left(\boldsymbol{\Theta}_m, \mathbf{p}_m; \hat{\mathbf{x}}_k\right)\right] \\
& \quad + \frac{1}{|\mathcal{S}_2|}\left[L_p\left(\boldsymbol{\Theta}, \mathbf{p}_m'; \hat{\mathbf{x}}_k\right) - L_p\left(\boldsymbol{\Theta}_m, \mathbf{p}_m'; \hat{\mathbf{x}}_k\right)\right] \\
& \quad + \frac{1}{|\mathcal{S}_3|}\left[L_q\left(\boldsymbol{\Theta}, \hat{\mathbf{p}}_l; \mathbf{q}_i\right) - L_q\left(\boldsymbol{\Theta}_{m,l}' \hat{\mathbf{p}}_l; \mathbf{q}_i'\right)\right] \\
& \quad + \frac{1}{|\mathcal{S}_3|}\left[L_q\left(\boldsymbol{\Theta}, \hat{\mathbf{p}}; \mathbf{q}_i\right) - L_q\left(\boldsymbol{\Theta}_m, \hat{\mathbf{p}}_l'; \mathbf{q}_i'\right)\right]
\end{aligned}
\tag{22}
$$

Considering Eq. (20) and Assumptions 1-3, we get

$$
\begin{aligned}
& B_L\left(\boldsymbol{\Theta}_m||\boldsymbol{\Theta}\right) + B_{L_m}\left(\boldsymbol{\Theta}||\boldsymbol{\Theta}_m\right) \\
& \leq \kappa\left(\frac{\sigma_p}{|\mathcal{S}_2|} + \frac{\sigma_{\boldsymbol{\Theta}}+1}{|\mathcal{S}_3|}\right)\left(\left\|\left(\boldsymbol{\Theta} - \boldsymbol{\Theta}_m\right)\mathbf{p}_m\right\|_2 + \left\|\left(\boldsymbol{\Theta} - \boldsymbol{\Theta}_m\right)\mathbf{p}_m'\right\|_2\right) \\
& \leq \kappa\left(\frac{\sigma_p}{|\mathcal{S}_2|} + \frac{\sigma_{\boldsymbol{\Theta}}+1}{|\mathcal{S}_3|}\right)\left(r\left\|\left(\boldsymbol{\Theta} - \boldsymbol{\Theta}_m\right)\mathbf{d}\right\|_2 + \frac{2\sigma_{\boldsymbol{\Theta}}\varepsilon}{|\Omega|}\right)
\end{aligned}
\tag{23}
$$

The last inequality in Eq. (23) holds with some mathematical manipulation of the reconstruction function shown in Assumption 3 and the inequality shown in Eq. (18).

**Proof Completing:** Let $U = \sum_{i=1}^{|\Omega|}\left\|\left(\boldsymbol{\Theta} - \boldsymbol{\Theta}_m\right)\mathbf{d}_i\right\|_2$, comparing the inequalities shown in Eq. (22) and Eq. (23), we can get

$$
\begin{aligned}
& \frac{2}{|\Omega|}\sum_{i=1}^{|\Omega|}\left\|\left(\boldsymbol{\Theta} - \boldsymbol{\Theta}_m\right)\mathbf{d}_i\right\|_2^2 \\
& \leq \kappa\left(\frac{\sigma_p}{|\mathcal{S}_2|} + \frac{\sigma_{\boldsymbol{\Theta}}+1}{|\mathcal{S}_3|}\right)\left(r\left\|\left(\boldsymbol{\Theta} - \boldsymbol{\Theta}_m\right)\mathbf{d}\right\|_2 + \frac{2\sigma_{\boldsymbol{\Theta}}\varepsilon}{|\Omega|}\right)
\end{aligned}
\tag{24}
$$

or equivalently,

$$
\frac{2}{|\Omega|}U^2 \leq \kappa\left(\frac{\sigma_p}{|\mathcal{S}_2|} + \frac{\sigma_{\boldsymbol{\Theta}}+1}{|\mathcal{S}_3|}\right)\left(rU + \frac{2\sigma_{\boldsymbol{\Theta}}\varepsilon}{|\Omega|}\right),
\tag{25}
$$

which can be further simplified by

$$
\begin{aligned}
U \leq & \frac{|\Omega|}{4}\kappa r\left(\frac{\sigma_p}{|\mathcal{S}_2|} + \frac{\sigma_{\boldsymbol{\Theta}}+1}{|\mathcal{S}_3|}\right) \\
& + \frac{|\Omega|}{4}\sqrt{\kappa^2 r^2\left(\frac{\sigma_p}{|\mathcal{S}_2|} + \frac{\sigma_{\boldsymbol{\Theta}}+1}{|\mathcal{S}_3|}\right)^2 - \frac{8\kappa\sigma_{\boldsymbol{\Theta}}\varepsilon}{|\Omega|^2}\left(\frac{\sigma_p}{|\mathcal{S}_2|} + \frac{\sigma_{\boldsymbol{\Theta}}+1}{|\mathcal{S}_3|}\right)}
\end{aligned}
\tag{26}
$$

Putting the above results into Eq. (16) gives

$$
\begin{aligned}
& \left|L_p\left(\boldsymbol{\Theta}, \mathbf{p}; \hat{\mathbf{x}}_k\right) - L_p\left(\boldsymbol{\Theta}_m, \mathbf{p}; \hat{\mathbf{x}}_k\right)\right| \\
& \leq \sigma_p\left\|\left(\boldsymbol{\Theta} - \boldsymbol{\Theta}_m\right)\mathbf{p}\right\|_2 \\
& \leq \sigma_p\left(r\left\|\left(\boldsymbol{\Theta} - \boldsymbol{\Theta}_m\right)\mathbf{d}\right\|_2 + \frac{2\sigma_{\boldsymbol{\Theta}}\varepsilon}{|\Omega|}\right) \\
& \leq \frac{|\Omega|\sigma_p r}{4}\left(\frac{\sigma_p}{|\mathcal{S}_2|} + \frac{\sigma_{\boldsymbol{\Theta}}+1}{|\mathcal{S}_3|}\right) + \sigma_{\boldsymbol{\Theta}}\sigma_p\varepsilon \\
& \quad + \frac{|\Omega|\sigma_p}{4}\sqrt{r^2\left(\frac{\sigma_p}{|\mathcal{S}_2|} + \frac{\sigma_{\boldsymbol{\Theta}}+1}{|\mathcal{S}_3|}\right)^2 + \left(\frac{\sigma_p}{|\mathcal{S}_2|} + \frac{\sigma_{\boldsymbol{\Theta}}+1}{|\mathcal{S}_3|}\right)\frac{8\sigma_{\boldsymbol{\Theta}}\varepsilon}{|\Omega|}}
\end{aligned}
\tag{27}
$$

which completes the proof of Theorem 1.    □

Theorem 1 shows that the generalization error of NSFP decreases with the reciprocal of the number of point clouds ($|\mathcal{S}_2|$ and $|\mathcal{S}_3|$), demonstrating its superior performance in the large-scale scene flow estimation (please see Tables 1 and 2), where $|\mathcal{S}_2| \to \infty$ and $|\mathcal{S}_3| \to \infty$, demonstrating the effectiveness of NSFP in the large-scale settings. We further provide the analysis for the MNSF method in the following.

**Remark 3.** *Theorem 1 establishes the bounded nature of the uniform stability of the NSFP, offering a fresh perspective on deriving stability properties for learning algorithms, including the NSFP and its various iterations.*

**Theorem 2.** *Let $\Theta_{\mathbf{fusion}} = \left[ \Theta_1^\top, \Theta_2^\top \right]^\top$ denote the parameters of the fusion model. For the proposed multi-frame scheme (MNSF), with high probability, its uniform stability ($\beta_{\mathrm{MNSF}}$) is bounded by*

$$\beta_{\mathrm{MNSF}} \leq \beta_{\mathrm{NSFP}} + O\left(\tfrac{1}{|\mathcal{S}_2|}\right), \tag{28}$$

*where $O\left(\tfrac{1}{|\mathcal{S}_2|}\right) = \frac{4\kappa^2 \sigma_{\mathcal{S}_3}^2}{\lambda |\mathcal{S}_2|} + \left( \frac{8\kappa^2 \sigma_{\mathcal{S}_3}^2}{\lambda} + 2\sigma_{\mathcal{S}_3} \right) \sqrt{\frac{\ln 1/\delta}{2|\mathcal{S}_2|}}$ and $\lambda = \frac{\|\Theta_2 \Theta_b\|_2^2}{\|\Theta_1 \Theta_f + \mathbf{I}\|_2^2}$. Variables $\kappa$, $\sigma_{\mathcal{S}_3}$, and $\delta$ can be considered as constants.*

*Proof.* With the theoretical results, we are ready to prove Theorem 2. Let $\Theta_{\mathbf{fusion}} = \left[ \Theta_1^\top, \Theta_2^\top \right]^\top$ denote the parameters of the fusion model. Considering a linear fusion function and inverter (defined by Eq. (12)), we have

$$\Theta \left[ \begin{array}{c} \mathbf{f} \\ \mathbf{f}' \end{array} \right] = \left[ \begin{array}{cc} \Theta_1 & \Theta_2 \end{array} \right] \left[ \begin{array}{c} \mathbf{f} \\ \mathbf{f}' \end{array} \right] = \left[ \begin{array}{cc} \Theta_1 & \Theta_2 \end{array} \right] \left[ \begin{array}{c} \Theta_f \mathbf{p} \\ -\Theta_b \mathbf{p} \end{array} \right] \tag{29}$$

Then, using Eq. (29), we can rewrite the loss function $L_p$ in MNSF optimization as

$$\begin{aligned} &\frac{1}{|\mathcal{S}_2|} \sum_{j=1}^{|\mathcal{S}_2|} \left\| (\Theta_1 \Theta_f - \Theta_2 \Theta_b) \, \mathbf{p}_j + \mathbf{p}_j - \hat{\mathbf{x}}_k \right\|_2^2 \\ &\leq \left\| \Theta_1 \Theta_f \mathbf{p}_j + \mathbf{p}_j - \hat{\mathbf{x}}_k \right\|_2^2 + \left\| \Theta_2 \Theta_b \mathbf{p}_j \right\|_2^2 \\ &= \left\| g\left( \mathbf{p} \right) - \hat{\mathbf{x}}_k \right\|_2^2 + \lambda \left\| g\left( \mathbf{p} \right) \right\|_2^2 \end{aligned} \tag{30}$$

where $\lambda = \frac{\|\Theta_2 \Theta_b\|_2^2}{\|\Theta_1 \Theta_f + \mathbf{I}\|_2^2}$. With Eq. (30) and Theorem 12 (Bousquet & Elisseeff, 2002), we finally obtain the theoretical results shown in Theorem 2.    □

**Remark 4.** *As demonstrated in Eq. (30), by employing an appropriate fusion strategy, our proposed MNSF emerges as a polynomial function of the approach utilized in NSFP, revealing a straightforward but essential variation of the NSFP algorithm.*

**Remark 5.** *Theorem 2 illustrates the advantageous impact of increasing the number of tasks for MNSF. To gain an intuitive grasp, let's examine an extreme scenario where all tasks (forward flow and backward flow optimization) are interconnected, each with an independently drawn sample size of one. Elevating the number of related tasks is akin to augmenting the independently drawn examples, undoubtedly aiding in the acquisition of related information. Theorem 2 substantiates this intuition with a theoretical assurance of rapid convergence rates comparable to those of NSFP.*

Theorem 2 reveals two key aspects of MNSF based on loss function in Eq. (5): 1) The algorithm's generalization error is inversely proportional to the number of point clouds, indicating its efficacy with large-scale point clouds (please see Tables 1 and 2); 2) Theoretical analysis shows that MNSF's generalization error upper bound is on par with NSFP's when $|\mathcal{S}_2| \to \infty$. This indicates that adding the $t$-1 frame into the optimization maintains and even enhances the generalization, as supported by the case studies.

## A.3 Additional experimental details and results

**Descriptions for dataset construction.** Following (Li et al., 2021; 2023), we first use the object information provided by Argoverse/Waymo to separate rigid and non-rigid segments. Then we extract the ground truth translation of rigid parts using the self-centered poses of autonomous vehicles and

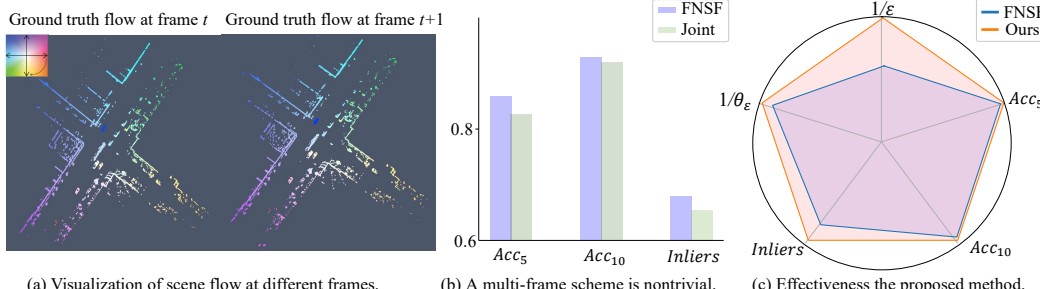

(a) Visualization of scene flow at different frames.    (b) A multi-frame scheme is nontrivial.    (c) Effectiveness the proposed method.

Figure 5: NSFP and FNSF show powerful generalization ability in large lidar autonomous driving scenes. However, none of these studies exploit the useful temporal information from previous point cloud frames. Extensive studies on optical flow estimation (Wulff et al., 2017; Golyanik et al., 2017; Janai et al., 2018; Maurer & Bruhn, 2018; Liu et al., 2019a; Stone et al., 2021; Hur & Roth, 2021; Mehl et al., 2023) and (a) have shown that scene flow in consecutive frames are similar to each other (*i.e.*, the upper left color wheel represents the flow magnitude and direction). To this end, an intuitive approach for exploiting temporal information, namely *Joint*, is to force a single FNSF to jointly estimate the previous flow ($t$-$1 \rightarrow t$) and the current flow ($t \rightarrow t$+1). (b) shows that such an intuitive multi-frame scheme achieves worse performance than two-frame FNSF on the Waymo Open dataset. In this paper, we propose a multi-frame point cloud scene flow estimation scheme. (c) shows that the proposed method achieves state-of-the-art on the Waymo Open dataset.

non-rigid parts using object poses, respectively. Thus, we can combine these translational vectors to generate the ground truth scene flow. Moreover, we remove the ground points using the information provided by the ground height map.

**Temporal encoding.** We also compare the proposed multi-frame scheme with the temporal encoding strategy, because temporal encoding is useful to process point cloud sequences (Wang et al., 2022a; Zheng et al., 2023). As aforementioned, it is difficult for FNSF (joint) to distinguish point clouds from different frames. To mitigate this issue, we use temporal encoding and concatenate the temporal coordinate into the spatial coordinate, *i.e.*, obtaining a 4D point cloud. In this way, we construct FNSF (temporal encoding) to jointly estimate the previous flow ($t$-$1 \rightarrow t$) and the current flow ($t \rightarrow t$+1). Table 1) and Table 2 show that FNSF (temporal encoding) slightly outperforms FNSF (joint). Such experimental result indicates that using temporal encoding partially addresses the issue in FNSF (joint) with limited performance improvement. However, FNSF (joint) is still *inferior to* the proposed method. The interpretation is that temporal encoding may be more suitable for long sequence point clouds than short sequence point clouds (Wang et al., 2022a). Therefore, the proposed method provides a promising solution to multi-frame point cloud scene flow estimation.

**Cycle consistency constraint.** We conduct experiments to figure out whether the proposed method can be further improved by the cycle/temporal consistency loss, because it is common practice to encourage the trajectory of point cloud to be smooth (Liu et al., 2019b; Mittal et al., 2020; Wang et al., 2022a) for multi-frame point clouds, by constraining the distance between point clouds from different frames. To this end, a temporal consistency loss or a cycle consistency loss is usually used during the training process of point cloud models. Table 1 and Table 2 show that adding the cycle consistency loss decreases the performance of the proposed method, *i.e.*, strict accuracy decreasing from 87.16/88.75% to 81.09/83.26%. In addition, the cycle consistency loss significantly increases the computational complexity, and the inference time costs 1831 ms. Thus, the cycle/temporal consistency loss is not necessary in our case. Such a finding also verifies the empirical observation in (Li et al., 2023). Therefore, we implicitly enforce cycle/temporal smoothness, instead of explicitly constraining cycle/temporal smoothness.

**Architecture of the temporal fusion model.** We provide results of MNSF with different architectures of the temporal fusion model. The temporal fusion model is an average operation, a learnable matrix $W$, and an MLP, respectively. Specifically, mean denotes directly computing the average of the forward and the inverted backward scene flow, *i.e.*, $(\mathbf{f} + \mathbf{f}')/2$. The weighted sum represents using the learnable matrix $W$ to adjust the weights between the forward and the inverted backward scene flow, *i.e.*, $W\mathbf{f} + (I - W)\mathbf{f}'$. In comparison, these two flows are concatenated as the input to

Table 6: **Performance of different architectures of the temporal fusion model on the Waymo Open dataset.** All compared methods are evaluated with the full point cloud as the input.

| Operation | $\mathcal{E}(m) \downarrow$ | $Acc_5(\%) \uparrow$ | $Acc_{10}(\%) \uparrow$ | $\theta_\epsilon(rad) \downarrow$ |
|---|---|---|---|---|
| Mean | 0.070 | 82.55 | 92.64 | 0.285 |
| Weighted sum | 0.097 | 84.18 | 92.42 | 0.286 |
| MLP | **0.066** | **87.16** | **93.39** | **0.273** |

the MLP, and the output is the fused flow. Table 6 shows that setting the temporal fusion model as an MLP achieves optimal performance.

**Number of frames.** We demonstrate the results of MNSF with different frame numbers. Specifically, we have point clouds from $t$-$(m$-$2)$, $\cdots$, $t$-$1$, $t$, and $t$+$1$ for the $m$-frame setting. We independently train $m$-$1$ models, predicting the forward flow $t \rightarrow t$+$1$ and $m$-$2$ backward flow $t \rightarrow t$-$1$, $t \rightarrow t$-$2$, $\cdots$, $t \rightarrow t$-$(m$-$2)$, respectively. Finally, we use a fusion model to estimate the final flow, *i.e.*, $t \rightarrow t$+$1$. Table 5 shows that the multi-frame setting outperforms the 2-frame setting. It verifies that exploiting temporal information from previous frames is useful for scene flow estimation. Table 5 also reveals that the contribution of the temporal information is incremental, when the number of frames is larger than three. Such a finding is consistent with the previous work in the optical flow estimation (Ren et al., 2019).

