# OpenReview forum: "Multi-Frame Neural Scene Flow: Learning Bounds and Algorithms"
_ICLR.cc/2025/Conference — ICLR 2025 Conference Withdrawn Submission_

### Official Review · Reviewer_defi · 2024-10-27

**Soundness:** 2
**Presentation:** 2
**Contribution:** 3
**Rating:** 5
**Confidence:** 3

**Summary:**

# Analysis of Multi-Frame Neural Scene Flow Paper

## Main Contributions and Methods

### 1. Theoretical Innovation
- First theoretical explanation of Neural Scene Flow Prior (NSFP)'s generalization capability
- Proved generalization bound is inversely proportional to point cloud size using uniform stability framework
- Established rigorous theoretical foundation for neural scene flow estimation

### 2. Technical Innovation
- Proposed Multi-frame Neural Scene Flow (MNSF) framework:
  * Utilizes three consecutive point cloud frames (t-1,t,t+1)
  * Employs dual FSNF models for forward/backward flow computation
  * Introduces temporal fusion module for multi-frame information integration
  * Proves similar generalization bounds as NSFP

## Critical Analysis

### 1. Application Concerns
The paper lacks clear justification for scene flow estimation in autonomous driving:
- Limited discussion of practical applications
- No clear connection to real-world autonomous driving needs (The author of the SCENE FLOW TASK conducted experiments on autonomous driving data sets, but the paper did not explain in detail and accurately what practical applications SCENE FLOW has. It seems that this task has no common applications in autonomous driving tasks, so Why do we need to continue to optimize the scene flow estimation model? And the algorithm of this paper was not originally designed to be tried and applied online in real cars?)

### 2. Performance-Efficiency Trade-off
Compared to FNSF baseline:
- Accuracy improvements:
  * EPE reduced by only 0.008
  * Marginal gains in other metrics
- Computational cost:
  * Processing time increased from 84ms to 160ms (8192 points)
  * Nearly 2x computational overhead for minimal accuracy gain

### 3. Theoretical-Empirical Inconsistency
- Best performance achieved with 3 frames
- Contradicts theoretical expectation that more frames should yield better results
- Needs explanation for diminishing returns with additional frames

### 4. Presentation Issues
- Over-emphasis on mathematical derivations
- Limited visualizations
- Suggests:
  * Move secondary equations to supplementary material
  * Add more visual explanations
  * Include ablation study visualizations

## Recommendations for Improvement
1. Strengthen practical relevance:
   - Clear application scenarios
   - Real-world performance requirements
   - Deployment considerations

2. Address efficiency concerns:
   - Optimize computational complexity
   - Explore lightweight alternatives
   - Justify accuracy-efficiency trade-off

3. Bridge theory-practice gap:
   - Explain optimal frame number
   - Analyze temporal correlation effects
   - Validate theoretical bounds empirically

4. Enhance presentation:
   - Streamline mathematical content
   - Add visual explanations
   - Include comprehensive ablation studies

**Strengths:**

### 1. Theoretical Innovation
- First theoretical explanation of Neural Scene Flow Prior (NSFP)'s generalization capability
- Proved generalization bound is inversely proportional to point cloud size using uniform stability framework
- Established rigorous theoretical foundation for neural scene flow estimation

### 2. Technical Innovation
- Proposed Multi-frame Neural Scene Flow (MNSF) framework:
  * Utilizes three consecutive point cloud frames (t-1,t,t+1)
  * Employs dual FSNF models for forward/backward flow computation
  * Introduces temporal fusion module for multi-frame information integration
  * Proves similar generalization bounds as NSFP

**Weaknesses:**

### 1. Application Concerns
The paper lacks clear justification for scene flow estimation in autonomous driving:
- Limited discussion of practical applications
- No clear connection to real-world autonomous driving needs (The author of the SCENE FLOW TASK conducted experiments on autonomous driving data sets, but the paper did not explain in detail and accurately what practical applications SCENE FLOW has. It seems that this task has no common applications in autonomous driving tasks, so Why do we need to continue to optimize the scene flow estimation model? And the algorithm of this paper was not originally designed to be tried and applied online in real cars?)

### 2. Performance-Efficiency Trade-off
Compared to FNSF baseline:
- Accuracy improvements:
  * EPE reduced by only 0.008
  * Marginal gains in other metrics
- Computational cost:
  * Processing time increased from 84ms to 160ms (8192 points)
  * Nearly 2x computational overhead for minimal accuracy gain

### 3. Theoretical-Empirical Inconsistency
- Best performance achieved with 3 frames
- Contradicts theoretical expectation that more frames should yield better results
- Needs explanation for diminishing returns with additional frames

### 4. Presentation Issues
- Over-emphasis on mathematical derivations
- Limited visualizations
- Suggests:
  * Move secondary equations to supplementary material
  * Add more visual explanations
  * Include ablation study visualizations

**Questions:**

Refer to the Summary section

---

### Official Review · Reviewer_Dnb9 · 2024-10-29

**Soundness:** 3
**Presentation:** 2
**Contribution:** 3
**Rating:** 6
**Confidence:** 3

**Summary:**

In this paper, the authors propose to estimate scene flow by using multiple frames. The authors first provide a theoretical analysis of the generalization error which shows that the upper bound of the generalization error can be reduced by using multiple frames. Further, for the first time, the authors propose a multi-frame scene flow optimization framework which is composed of two forward models, an inverse model, and a temporal fusion model. Experimental results show that the proposed MNSF can achieve SOTA performance on multiple datasets (*e.g.*, Waymo and Argoverse) when using both full point clouds and sampled point clouds.

**Strengths:**

- For the first time, the authors propose to estimate scene flow by using multi-frame information which is reasonable.
- The authors analyze the upper bound of the generalization error in both NSFP and MNSF settings.
- Experimental results are consistent with the theory.

**Weaknesses:**

- First, I wonder if the proposed method and the theoretical analysis of the upper bound of the generalization error can be generalized to more frames such as $x_{t-2}, x_{t-1}, x_t$ or more.
- Why cannot use two identical forward models instead of using a single forward model? Does this mean we need further add more forward models when using more frames?
- I think some related works may be ignored. The forward and backward accumulation has been used in multi-frame optical flow field such as [A][B]. The authors should consider reviewing this kind of work.
- Why does the performance slightly decrease when just using $g_{fusion}$?

[A] Shi X, Huang Z, Bian W, et al. Videoflow: Exploiting temporal cues for multi-frame optical flow estimation[C]//Proceedings of the IEEE/CVF International Conference on Computer Vision. 2023: 12469-12480.
[B] Wu G, Liu X, Luo K, et al. Accflow: Backward accumulation for long-range optical flow[C]//Proceedings of the IEEE/CVF International Conference on Computer Vision. 2023: 12119-12128.

**Questions:**

Please refer to the paper weakness.

---

### Official Review · Reviewer_PZN8 · 2024-11-04

**Soundness:** 2
**Presentation:** 2
**Contribution:** 2
**Rating:** 6
**Confidence:** 3

**Summary:**

This paper introduces Multi-Frame Neural Scene Flow (MNSF), an extension of Neural Scene Flow Prior (NSFP), aimed at enhancing 3D scene flow estimation for autonomous driving. By incorporating multi-frame temporal data, MNSF achieves better temporal consistency and outperforms NSFP and FNSF, especially in large-scale environments. Eexperimental results on the Waymo Open and Argoverse datasets verify its robust performance, even in fast-motion scenarios.

**Strengths:**

The overall presentation is good, and figures are informative.
Unfortunately, I cannot understand the importance of this task, but the methodology is very sounding.
The increased generalizability and robustness to the fast motion is impressive.

**Weaknesses:**

1. I think the introduction section should include more detail about the task itself. Why should we solve NSFP? What is the application of this task?

2. It seems that the flow inversion can be done without neural network. Please justify the design of temporal scene flow inversion module (g_invert). And if possible, it would be good to check the results of the models individually trained with 1) forward branch and 2) backward+inversion branch. Please show the quan/qual results.

3. Why the proposed method can handle fast motion? Please discuss more about it.

**Questions:**

See the weaknesses.

---

### Official Review · Reviewer_oAqm · 2024-11-04

**Soundness:** 2
**Presentation:** 2
**Contribution:** 2
**Rating:** 3
**Confidence:** 3

**Summary:**

The work explores the scene flow estimation using multi-frame information (specifically, three point cloud frames), advancing on the earlier FSNF/NSFP approach. It includes a theoretical analysis of generalization capability via uniform stability.

**Strengths:**

The idea of MNSF, which uses networks to optimize the fusion of forward and backward flow, is good, straightforward, and easy to understand.

**Weaknesses:**

-	Incomplete literature study, leading to repeated incorrect key statements and incomplete benchmarks.

(a) one major critique of the data-driven method presented in this paper is that it cannot handle large-scale datasets, such as those with all points in the Argoverse and Waymo datasets (which can exceed 10,000 points/frame). However, this assertion is inaccurate, as demonstrated by data-driven methods like FastFlow3D [1], ZeroFlow [2], Deflow [3], TrackFlow [4], Seflow [5], and Flow4D [6], which can process these datasets without needing to downsample the input points. [1,2] are cited but not compared. Others are not cited or compared.

(b) The absence of a discussion on other works that use multi-frame. Specifically, approaches such as TrackFlow [4] and Flow4D [6], are not mentioned.

-	The metrics used for the scene flow estimation are outdated and need to be updated. According to the AV scene flow leaderboard [7], recent metrics emphasize the accuracy of flow for dynamic points, using average EPE for static and dynamic points, rather than averaging errors across all points. Since the portion of dynamic points among all points is very small, ~ 10-20%, and it is important to capture motion in autonomous driving scenarios, it is important to include the EPE for dynamic metrics.
-	I have not reviewed the mathematical derivations in detail. However, regarding the conclusion from theoretical proofs that an increased number of points leads to better generalization and lower error rates, the experimental results shown in Table 5 do not seem to support this theory (?). When the number of frames increases from 2 to 5, no incremental performance improvement is observed.

---------------------------------------------------------------------------


[1] FastFlow3D: Jund, Philipp, et al. "Scalable scene flow from point clouds in the real world." IEEE Robotics and Automation Letters 7.2 (2021): 1589-1596.

[2] Zeroflow: Vedder, Kyle, et al. "ZeroFlow: Scalable Scene Flow via Distillation." 2024 ICLR.

[3] Deflow: Zhang, Qingwen, et al. "DeFlow: Decoder of scene flow network in autonomous driving." 2024 ICRA.

[4] TrackFlow: Khatri, Ishan, et al. "I Can’t Believe It’s Not Scene Flow. 2024 ECCV.

[5] Seflow: Zhang, Qingwen, et al. "SeFlow: A Self-supervised Scene Flow Method in Autonomous Driving." 2024 ECCV.

[6] Flow4D: Kim, Jaeyeul, et al. "Flow4D: Leveraging 4D Voxel Network for LiDAR Scene Flow Estimation." arXiv preprint arXiv:2407.07995 (2024).

[7] https://eval.ai/web/challenges/challenge-page/2210/leaderboard/5463

**Questions:**

1.	Line 241, for the inverse model, do you introduce a negative sign to the backward flow, or is this achieved through network optimization? If the latter, why not simply add a negative sign?
2.	Figure 2 is unclear. In the zoom-in view, could you explain what the light blue and orange colors represent?
3.	Line 406, this part claims MFSN has better OOD generalizability. What is the OOD data used in the experiments? It is unclear how Tables 1 and 2 show the OOD generalizability.
4.	In Fig 4, it appears that MNFS fails to detect the dynamic flow of smaller objects, such as a pedestrian walking beside the street in front of a building, compared with FNSF. Is detecting smaller moving objects a challenge for MNFS?

---

### Note · Authors · 2024-11-25

**Comment:**

We would like to appreciate the reviewers' efforts and the valuable suggestions. We will improve the paper and address all issues accordingly in the next submission.

**Withdrawal Confirmation:**

I have read and agree with the venue's withdrawal policy on behalf of myself and my co-authors.